

# Odor-dependent temporal dynamics in *Caenorhabitis elegans* adaptation and aversive learning behavior

Jae Im Choi[1,*], Hee Kyung Lee[1,2,*], Hae Su Kim[1,*], So Young Park[1,*], Tong Young Lee[1], Kyoung-hye Yoon[1,2] and Jin I. Lee[1]

[1] Division of Biological Science and Technology, Yonsei University, Wonju, Gangwondo, South Korea
[2] Mitohormesis Research Center, Department of Physiology, Wonju College of Medicine, Yonsei University, Wonju, Gangwondo, South Korea
[*] These authors contributed equally to this work.

## ABSTRACT

Animals sense an enormous number of cues in their environments, and, over time, can form learned associations and memories with some of these. The nervous system remarkably maintains the specificity of learning and memory to each of the cues. Here we asked whether the nematode *Caenorhabditis elegans* adjusts the temporal dynamics of adaptation and aversive learning depending on the specific odor sensed. *C. elegans* senses a multitude of odors, and adaptation and learned associations to many of these odors requires activity of the cGMP-dependent protein kinase EGL-4 in the AWC sensory neuron. We identified a panel of 17 attractive odors, some of which have not been tested before, and determined that the majority of these odors require the AWC primary sensory neuron for sensation. We then devised a novel assay to assess odor behavior over time for a single population of animals. We used this assay to evaluate the temporal dynamics of adaptation and aversive learning to 13 odors and find that behavior change occurs early in some odors and later in others. We then examined EGL-4 localization in early-trending and late-trending odors over time. We found that the timing of these behavior changes correlated with the timing of nuclear accumulation of EGL-4 in the AWC neuron suggesting that temporal changes in behavior may be mediated by aversive learning mechanisms. We demonstrate that temporal dynamics of adaptation and aversive learning in *C. elegans* can be used as a model to study the timing of memory formation to different sensory cues.

## INTRODUCTION

The ability to form associations to odors is inherent to most animals. In particular, the formation of memories to biologically pertinent odors can be important for survival and reproduction. For instance, mother sheep learn the particular odors of their own lambs rather than other stranger lambs within hours of birth (*Broad et al., 2002*). This specific odor memory formation will be important for the mother sheep to establish bonding behaviors with her young (*Levy, Keller & Poindron, 2004*).

Corresponding authors
Kyoung-hye Yoon,
kyounghyeyoon@yonsei.ac.kr
Jin I. Lee, jinillee@yonsei.ac.kr

The nematode *Caenorhabditis elegans* is attracted to dozens or more odors, and attraction is mediated by two pairs of olfactory sensory neurons, the AWC and the AWA neurons (*Bargmann, Hartwieg & Horvitz, 1993*). Persistent odor stimulation in the absence of food results in a decreased attraction to the odor (*Bargmann, Hartwieg & Horvitz, 1993*; *Colbert & Bargmann, 1995*; *Ward, 1973*). After 30 min of odor exposure to AWC-sensed odors, *C. elegans* displays a weak and transient decrease in attraction called short-term adaptation (henceforth called "adaptation") that requires the cyclic nucleotide-gated channel subunit CNG-3 (*O'Halloran et al., 2017*) and phosphorylation of the cyclic nucleotide-gated channel subunit TAX-2 by the cGMP-dependent protein kinase EGL-4 (*L'Etoile et al., 2002*).

Adaptation is not the only change that occurs upon a persistent odor: after an hour of odor exposure, an association between odor and food deprivation forms that requires insulin signaling and nuclear localization of EGL-4 in the AWC neuron (*Cho et al., 2016*; *L'Etoile et al., 2002*; *Lee et al., 2010*). This aversive olfactory learning results in a stable and long-lasting behavior change towards the specific odor (*Lee et al., 2010*).

Adaptation and aversive learning have been studied for the AWC-sensed odor benzaldehyde and butanone (*Cho et al., 2016*; *Colbert & Bargmann, 1995*; *L'Etoile et al., 2002*; *Lee et al., 2010*). However, the temporal dynamics of adaptation and aversive learning to other odors have not been tested thoroughly, and it is unknown whether associations to certain odors can form faster than to other odors. The standard odor treatment and chemotaxis assay protocol to measure changes in *C. elegans* odor behavior is a robust assay that has led to the discovery of odor adaptation and odor learning, and revealed the molecular pathways involved (*Bargmann, Hartwieg & Horvitz, 1993*; *Cho et al., 2016*; *Colbert & Bargmann, 1995*; *L'Etoile & Bargmann, 2000*; *L'Etoile et al., 2002*; *Lin et al., 2010*; *O'Halloran et al., 2009*). However, for a thorough temporal characterization of adaptation and associative learning to multiple odors, the standard assay would require a separate behavior assay for each time point for each odor, necessitating an inordinately large amount of worms and resources for such a study.

In this study we sought to investigate the temporal dynamics of aversive learning to a larger panel of odors by designing a new odor behavior assay called the real-time odor behavior assay. We show that time-dependent behavior changes due to odor adaptation and aversive learning can be observed in this novel assay. We then used the real-time behavior assay to test the temporal dynamics of odor behavior to a panel of 13 total odors. We found the timing of behavior change was specific for each odor, and temporal dynamics were correlated to the localization of EGL-4 in the AWC neuron.

## MATERIALS AND METHODS

### Nematode culture and strains

Worms were grown and maintained at 20 °C on Nematode Growth Medium (NGM) plates seeded with E. coli OP50 as described previously (*Brenner, 1974*). Strains used for this study, N2, *ceh-36, odr-7, pyIs500* ((p)*odr-3*::GFP::*egl-4*), were obtained from the *Caenorhabditis* Genetic Center (University of Minnesota, USA).

**Table 1  AWC and AWA primary sensory neurons primary sensory neurons mediate the attraction to 17 odors.** Wild-type indicates N2 strain, AWC- indicates the ceh-36 mutant strain, AWA- indicates odr-7 mutant strain. The right column indicates the neuron(s) responsible for sensing each odor. Extra large dot = 0.6–1.0 attraction index (AI), large dot = 0.4–0.6 AI, medium dot = 0.2–0.4 AI, and small dot < 0.2 AI.

| Odor | Wild-type | AWC- | AWA- | AWA/AWC |
|------|-----------|------|------|---------|
| diacetyl | ● | ● | • | AWA |
| benzaldehyde | ● | • | ● | AWC |
| butanone | ● | • | ● | AWC |
| isobutyric acid | ● | ● | • | AWA |
| 2-isobutylthiazole | ● | ● | ● | AWC |
| dimethylthiazole | ● | ● | ● | AWC |
| 2,4,5-trimethylthiazole | ● | ● | ● | AWA or AWC |
| 2-methylpyrazine | ● | • | ● | AWC |
| 2-heptanone | ● | • | ● | AWC |
| 1-methylpyrrole | ● | • | ● | AWC |
| 4-chlorobenzyl mercaptan | ● | • | ● | AWC |
| butyric acid | ● | ● | • | AWA |
| 1-pentanol | ● | • | ● | AWC |
| benzyl mercaptan | ● | • | ● | AWC |
| 2-cyclohexylethanol | ● | • | ● | AWC |
| 2-ethoxythiazole | ● | • | ● | AWC |
| benzyl proprionate | ● | ● | • | AWA and AWC |

## Behavior assays

A rough screen for behavioral attraction to a large set of odor chemicals was first conducted which identified dozens of attractive odors. From this smaller group, a subset of 17 highly attractive odors were tested for chemotaxis in wild-type and mutant strain worms, among which were many odors that have not been tested previously. Standard odor chemotaxis assays were carried out using previously established protocols with a few changes (*Bargmann, Hartwieg & Horvitz, 1993*). The following 17 odors in Table 1 were used: benzaldehyde, butanone, isoamyl alcohol (Sigma-Aldrich, St. Louis, MO,

USA); isobutyric acid, 2-isobutyl thiazole, dimethylthiazole, 2,4,5-trimethylthiazole, 2-methylpyrazine, 2-heptanone, 4-chlorobenzyl mercaptan, butyric acid, 1-pentanol, benzyl mercaptan, 2-cyclohexylethanol, benzyl proprionate (Alfa Aesar, South Korea); 1-methylpyrrole, 2-ethoxythiazole (TCI, Tokyo, Japan); diacetyl (Acros Organics, Geel, Belgium). All odors were diluted from original stock to a 1:100 dilution with ethanol except for benzaldehyde (1:200) and 2,4,5-trimethylthiazole (1:1,000). A total of 4 μl of diluted odor or ethanol was placed on the assay plate as attractant and counterattractant, respectively. A total of 2 μl of $NaN_3$ was placed with the attractant and counterattractant. Adult worms were washed in S-basal buffer (5.85 g NaCl, 1 g $K_2HPO_4$, 6 g $KH_2PO_4$) three times, placed on the assay plate, and allowed to move freely for at least one hour before counting. We set a threshold for whether a neuron is required for detection as an AI in either *ceh-36* or *odr-7* mutants that was less than 50% of the wild-type AI.

Standard odor memory behavior assays were carried out similar to previous published protocols (*Colbert & Bargmann, 1995*; *L'Etoile & Bargmann, 2000*). Before assaying behavior, worms were washed in S-basal buffer three times, and placed in 1 ml of benzaldehyde or 2,4,5-TMT diluted at 1:10,000 in S-basal buffer for 0–120 min at 10 min or 20 min intervals. After odor exposure, worms were washed three times in S-basal and behavior was assayed.

## Real-time odor behavior assay

Media for the assay plate is as follows: 1.6% Difco Granulated Agar (Beckton Dickinson, Franklin Lakes, NJ, USA) was dissolved in water by heating, and 1mM of $CaCl_2$, 1mM of $MgSO_4$ and 5 mM of $KPO_4$ buffer (108.3 g KH2PO4, 35.6 g K2HPO4, H2O to 1 liter) was mixed with the agar. Media was dispensed into 12.5 cm × 12.5 cm × 2 cm square plastic culture plates (SPL Lifescience, Pocheon, South Korea), and allowed to harden for at least 3 h. Odors were diluted with ethanol to the following concentrations: benzaldehyde (1:100), butanone (1:100), diacetyl (1:100), isoamyl alcohol (1:1,000), 2-heptanone (1:1,000), 2,4,5-TMT (1:500), 4-chlorobenzyl mercaptan (1:500), 2-ethoxythiazole (1:300), 2-cyclohexylethanol (1:5,000), 1-methylpyrrole (1:500), 1-pentanol (1:10,000), 2-isobutylthiazole (1:100), 2-methylpyrazine (1:10). Diluted attractant odors were placed in the middle of the ''A'' section of the assay plate (Fig. 1), and a dab of OP50 strain *E. coli* bacteria from a standard nematode growth media plate with a platinum worm picker was placed in the middle of the ''F'' section of the assay plate as a counterattractant, no larger than 3 mm diameter. The counterattractant itself is a very weak attractant (Fig. 1B, control), and overall aids this assay in resolving the attenuation of odor attraction after memory has formed. Washed worms are then placed in the middle of the plate, and dried by wicking the buffer using an unscented tissue. The plates were tapped somewhat vigorously which aids in worm movement away from the center of the plate. Worms were counted in each section A, B, E, and F every ten minutes for 120 min and AI was calculated ($AI = [[2(\#$ of worms in A) $+ (\#$ of worms in B)] $- [(\#$ of worms in E) $+ 2(\#$ of worms in F)]]/2(\#$ of worms in $A + B + E + F$)). 50% total behavior change was calculated as (30 min AI $-$ 120 min AI)/2, and time to 50% behavior change for each odor and each trial was determined.

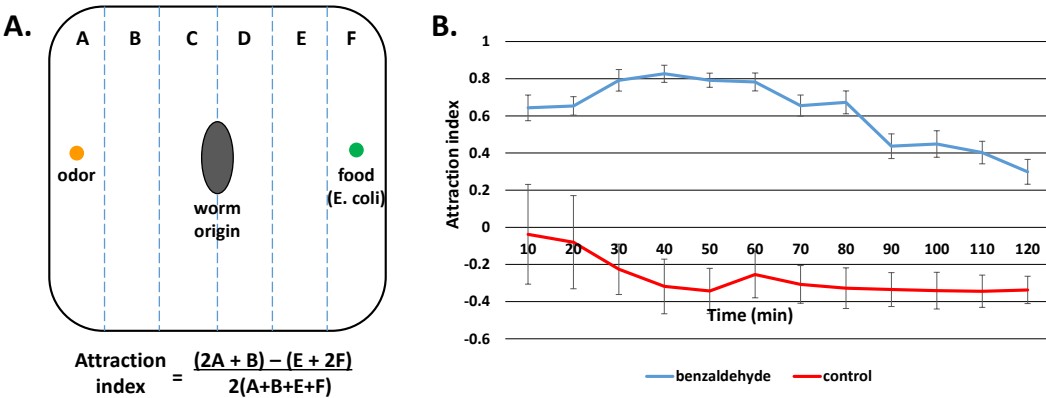

**Figure 1** **Real-time odor behavior assay.** (A) Assay design. Worms are placed in the middle of a 15 cm square plate. Odor solution is placed on one side of the plate, and a weak counterattractant (*E. coli* strain OP50) is placed on the opposite side. (B) Benzaldehyde attraction in real-time odor behavior assay. Attraction to the odor benzaldehyde (blue) decreases over the two hour assay. Control diluent (100% ethanol) attraction is shown in red.

## Quantification of GFP::EGL-4 localization

Subcellular localization of GFP::EGL-4 in *py* Is500 ((p)*odr-3*::GFP::EGL-4; (p)*odr-1*::RFP) integrated strain worms during odor memory formation was performed as previously published (*Lee et al., 2010*). Briefly, *py* Is500 animals were exposed to odor dilutions of either benzaldehyde (1:10,000), 2-ethoxythiazole (1:10,000), 1-methylpyrrole (1:10,000), 4-chlorobenzyl mercaptan (1:5,000), or 2,4,5-TMT (1:10,000) in S-basal for 0–120 min at 20 min intervals. After odor exposure, worms were placed on slides containing a 2% agar dried pad with 1 μl of 1M $NaN_3$ added to anesthetize the animals and observed under a fluorescent microscope. 20 animals were counted per sample, and EGL-4 localization was determined as either cytoplasmic or nuclear in the AWC neuron for each animal. In the rare case when localization in an animal could not be categorized as either nuclear or cytoplasmic, we would count that worm as nuclear, and then counted an extra animal. The next time this occurred, we counted that worm as cytoplasmic, and counted an extra animal. After the data was plotted, a regression equation was determined (Microsoft Excel) for each odor as follows: benzaldehyde ($y = -0.6424 \times 3 + 7.3661 \times 2 - 11.992 \times +13.304$), 2-ethoxythiazole ($y = -0.8924 \times 3 + 9.7545 \times 2 - 16.514 \times +19.071$), 1-methylpyrrole ($y = -0.9833 \times 3 + 12.252 \times 2 - 32.264 \times +31.657$), 4-chlorobenzyl mercaptan ($y = -0.4456 \times 3 + 6.1379 \times 2 - 14.315 \times +17.5$), 2,4,5-TMT ($y = -0.3003 \times 3 + 3.4079 \times 2 - 1.5239 \times +5.8048$). Fit and statistical significance of the regressions were determined by calculating Pearson's correlation coefficient and ANOVA analysis. Equations were used to calculate the approximate time at which 50% of animals display nuclear EGL-4.

# RESULTS

## Attraction to novel odors mediated by the AWC or the AWA primary sensory neurons

In order to test adaptation and aversive learning to a larger panel of odors than has been previously tested, we first identified 17 highly attractive odors (see Methods), among which were many odors that have not been tested previously (Table 1, Fig. S1). To identify the neuron responsible for sensing each odor, we subjected wild-type and olfactory sensory neuron mutants to these odors in a standard chemotaxis assay (*Bargmann, Hartwieg & Horvitz, 1993*; *Lanjuin et al., 2003*; *Sengupta, Colbert & Bargmann, 1994*). As predicted, diacetyl attraction is lost in the AWA- *odr-7* mutants and maintained in AWC-*ceh-36* mutants (Table 1; Fig. S1). On the contrary, benzaldehyde attraction disappears in the AWC- *ceh-36* mutants but is normal in AWA- *odr-7* mutants. Finally, 2,4,5-trimethylthiazole which is sensed by both AWA and AWC neurons (*Bargmann, Hartwieg & Horvitz, 1993*) remains attractive in both AWC- and AWA- mutants. When we tested the other odors, we found that attraction to most of the odors required the AWC sensory neurons, with only butyric acid and isobutyric acid sensed by the AWA neuron (Table 1; Fig. S1). Benzyl proprionate attraction was effectively lost in both AWC- and AWA- mutants. Hence, we have identified many novel attractive AWC-sensed odors.

## Odor adaptation and aversive learning in the real-time odor behavior assay

Adaptation and aversive learning have been studied for the AWC-sensed odor benzaldehyde (*Colbert & Bargmann, 1995*; *L'Etoile et al., 2002*; *Lee et al., 2010*). However, the temporal dynamics of aversive learning and memory to other odors have not been tested thoroughly. Therefore, we sought to characterize the change in odor attraction over time to a multitude of AWC-sensed odors.

Due to the limitations of the standard odor assay, we designed a new odor behavior assay we call the "real-time odor behavior assay" described in the Methods (Fig. 1A), in which worms are tracked every 10 min as they move towards or away from the odor in a two-hour time frame. When tested with benzaldehyde, worms spent the first 30–40 min moving towards the odor, which was demonstrated by the increase in attraction index (AI). AI remained high for another 30 min, until it started to decrease at 60 min and continued to fall during the remainder of the assay (Fig. 1B). In contrast, worms given only the diluent ethanol showed no such patterns of movement and their attraction index remained steady throughout the assay (Fig. 1B). The pattern of initial attraction followed by decrease in attraction was similar to what is observed in the standard assay (Fig. 2A). To make a side-by-side comparison with the standard assay, we took into consideration that both the tracking of worm movement and odor exposure occur concurrently with the assay. Therefore, we made the 30-minute time point to be the "beginning" of the real-time assay. When compared this way, similar patterns of overall decrease in attraction could be observed in both assays.

To test whether this assay could indeed be used to assess adaptation and aversive learning to an odor, we took advantage of mutants that are defective in each of these odor behaviors.

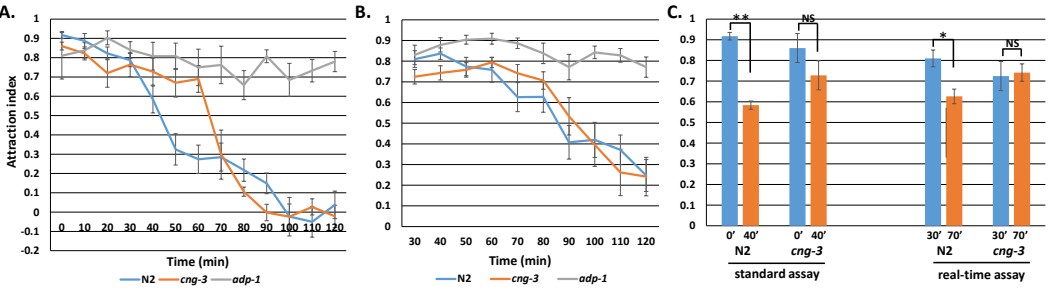

**Figure 2 Short-term adaptation and aversive learning to the odor benzaldehyde in the real-time odor behavior assay.** Attraction over time with N2 wild type, *cng-3* short-term adaptation defective mutants, and *adp-1* aversive learning defective mutants in the standard behavior assay (A) and the real-time odor behavior assay (B). (C) Short-term odor adaptation in the standard and real-time behavior assays. Comparison of the 0′ and 40′ minute time points in (A) with the 30′ and 70′ min time points in (B) for N2 and *cng-3* mutants. Error bars indicate standard error. Significance calculated by student's *T*-test. * indicates $p < 0.05$, ** indicates $p < 0.001$, NS indicates not significant.

In the traditional assay, adaptation-defective *cng-3* mutants showed high attraction for the first 60 min followed by a precipitous drop, demonstrating selective deficit for adaptation but not aversive learning (Fig. 2A). *cng-3* worms showed a similar pattern in the real-time odor assay, where decline in attraction index occurred at a later time point compared to wild type, although the difference did not seem as drastic at first glance. However, when we compared attraction at the 0 and 40 min time points in the standard assay to the 30 and 70 min time points in the real-time assay, the decrease in attraction seen in wild-type N2 animals was absent in *cng-3* mutants for both the standard and real-time assays (Fig. 2C).

Next, we looked at whether aversive learning could be assessed in the real time assay by using the aversive learning-defective *adp-1* mutant. Consistent with previous studies, *adp-1* mutants displayed no change in attraction even after a 120 min odor exposure in the standard assay (*Colbert & Bargmann, 1995*; Fig. 2A). Likewise, in the real time assay, *adp-1* mutants maintained high attraction throughout the course of the assay (Fig. 2B). Thus, both adaptation and aversive learning can be resolved in the real-time odor behavior assay.

## Timing of adaptation and aversive learning varies by odor

Since we demonstrated that our assay can accurately assess the timing of benzaldehyde adaptation and aversive learning, we conducted a comprehensive temporal analysis of behavior change to a palette of 12 other odors, nine of which have never been tested for odor adaptation or learning. These include alcohols, ketones, thiols, thiazoles, aromatics, and pyrazines. 30 min into the assay when we start the analysis, we observe differences in attraction to each odor ranging from 0.91 AI (2,4,5-trimethylthiazole) to 0.50 AI (2-heptanone). However, over time, the total decrease in attraction to all of the odors is similar over the entire assay (average AI decrease $= -0.475 \pm 0.025$) with one exception, the odor cyclohexylethanol, which displays only a modest attenuation (AI decrease $= -0.19$).

In all the odors tested, one can observe a distinct pattern of change in worm behavior towards these twelve odors over time. For AWC-sensed odors, behavior change generally occurs relatively slowly at the beginning of the assay then proceeds more quickly towards

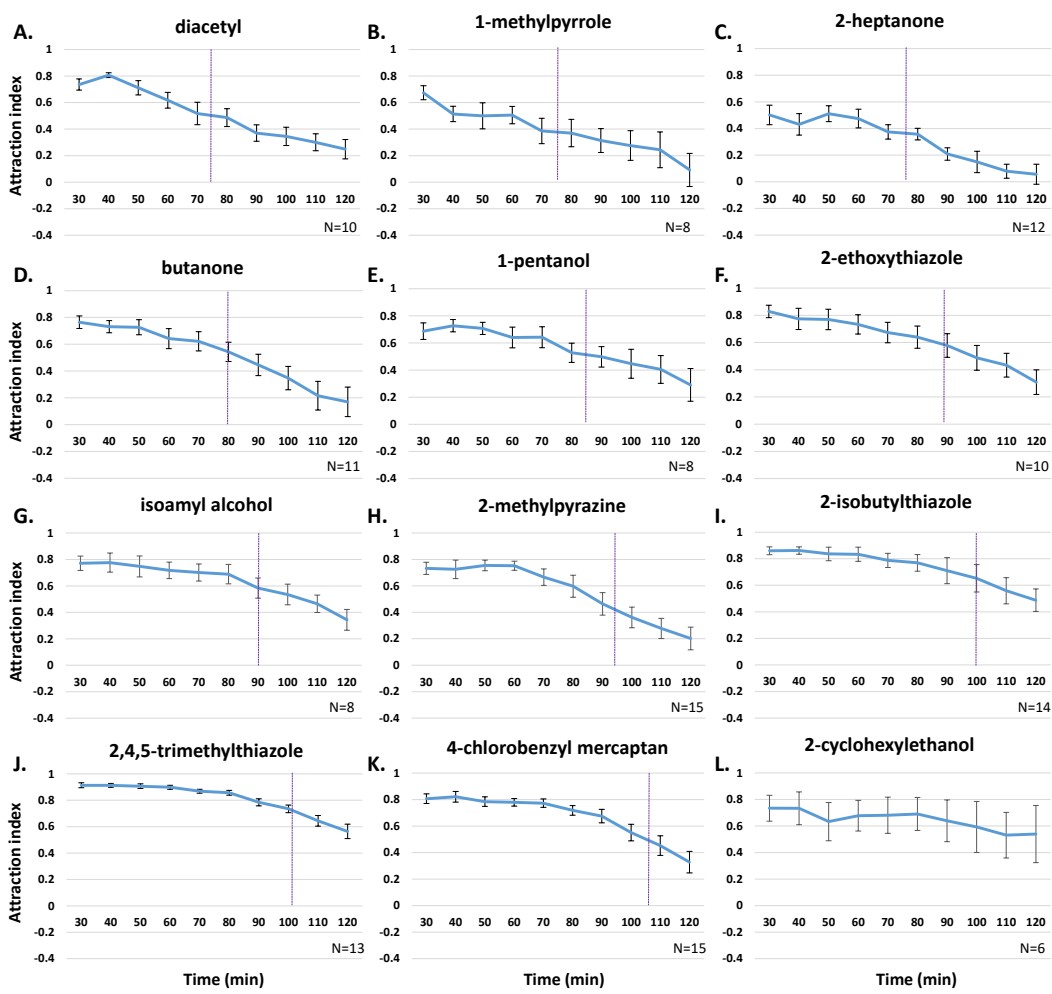

**Figure 3** **Temporal dynamics of odor learning to 12 odors.** (A) Diacetyl; (B) 1-methylpyrrole; (C) 2-heptanone; (D) butanone; (E) 1-pentanol; (F) 2-ethoxythiazole; (G) isoamyl alcohol; (H) 2-methylpyrazine; (I) 2-isobutylthiazole; (J) 2,4,5-trimethylthiazole; (K) 4-chlorobenzyl mercaptan; (L) 2-cyclohexylethanol. Purple dotted line marks approximate place where 50% of total behavior change occurs for each odor. Error bars indicate standard error.

the end (Fig. 3). Interestingly, for the odor diacetyl, which is the only odor here not sensed by the AWC neuron, behavior change appears to arise early and fairly consistently during the entire assay (Fig. 3).

Although overall changes in odor behavior patterns over the 120 min assay is similar for most of the odors tested, the timing of adaptation and aversive learning varies depending on the odor. For instance, benzaldehyde attraction decreases overall from 0.79 AI to 0.27 from 30 to 120 min, and 50% of this behavior change occurs at about 80 min (Fig. 3; Fig. S2). We then calculated the time at which 50% behavior change occurs for each of the odors (Fig. 3; Fig. S2A), and observed early trends in behavior change in diacetyl attraction (76.0 min) and 1-methylpyrrole attraction (76.3 min), as well as other odors (Fig. 3, Fig. S2A). On the other hand, late trends in behavior change were observed in

2,4,5-trimethylthiazole (2,4,5-TMT) attraction (100.0 min) and 4-chlorobenzyl mercaptan (4-CB) attraction (105.5 min) (Fig. 3, Fig. S2A). To confirm whether the late change in behavior was actually a result of aversive learning, we tested the attraction of *adp-1* mutants to 4-CB and 2,4,5-TMT attraction in the real-time assay (Fig. S3). Wild-type animals showed large decreases in attraction over the whole assay for both 4-CB (change in $AI = -0.478$) and 2,4,5-TMT (change in $AI = -0.349$). However, the change in behavior over time was minimal in *adp-1* mutants for both 4-CB attraction (change in $AI = -0.226$) and 2,4,5-TMT attraction (change in $AI = -0.009$). Thus, the late trend in decreased odor attraction for the two odors is likely due to late odor learning. We also confirmed that the early trend in benzaldehyde associative learning and the late trend in 2,4,5-TMT learning can be observed using either the real-time assay or the standard assay (Fig. S4). Thus, temporal dynamics of adaptation and aversive learning varies depending on the specific odor.

One possibility that may explain the variance in odor-dependent behavior changes over time is the differences in attractive strength for each odor. Odors elicit differential attraction in the standard assay (Table 1) and in our real-time behavior assay (Fig. 3) when worms first encounter the odors. This can be due to both inherent qualities of the odor and odor concentration (*Bargmann, Hartwieg & Horvitz, 1993*). To investigate the relationship between the initial attraction and subsequent behavior changes to the odor, we first grouped odors into either "late-trending" odors in which behavior change occurs late and "early-trending" odors in which behavior change occurs earlier (Fig. S3). The three late-trending odors, 4-CB, 2,4,5-TMT, isobutylthiazole, had an average initial AI of $0.857 \pm 0.045$, which was higher than the initial AI of the early-trending odors (average $AI = 0.721 \pm .098$; Fig. S5). However, the difference between initial AI of late and early-trending odors was not significant (Fig. S5).

We next wondered whether animals that initially displayed a low attraction to late-trending odors could change their behavior toward the odor faster than animals that initially displayed a high attraction to the same concentration of odor. However, animals that had an initially low AI to 4-CB (average $AI = 0.71 \pm .088$) and 2,4,5-TMT (average $AI = 0.81 \pm .043$) did not display faster behavior changes than animals that had an initially high AI (Fig. S6). Therefore, we did not find a significant relationship between temporal patterns of behavior change and initial attraction to the odors.

### Timing of aversive learning formation correlates with the timing of EGL-4 nuclear localization

Odor adaptation and aversive learning to AWC-sensed odors requires EGL-4 activity: cytoplasmic EGL-4 activity directs short-term adaptation whereas aversive learning requires EGL-4 to translocate to the nucleus and phosphorylate nuclear targets (*Juang et al., 2013*; *L'Etoile et al., 2002*). Because we showed that behavior changes over time in response to both benzaldehyde and 2,4,5-TMT require *adp-1* (Fig. 2B, Fig. S3), we wondered whether these changes in odor attraction were correlated to the sub-cellular localization of a functional GFP-tagged EGL-4 in the AWC neuron (Fig. 4A). We narrowed down our pool of odors to 5 AWC-sensed odors that induced consistent changes in behavior and grouped

 

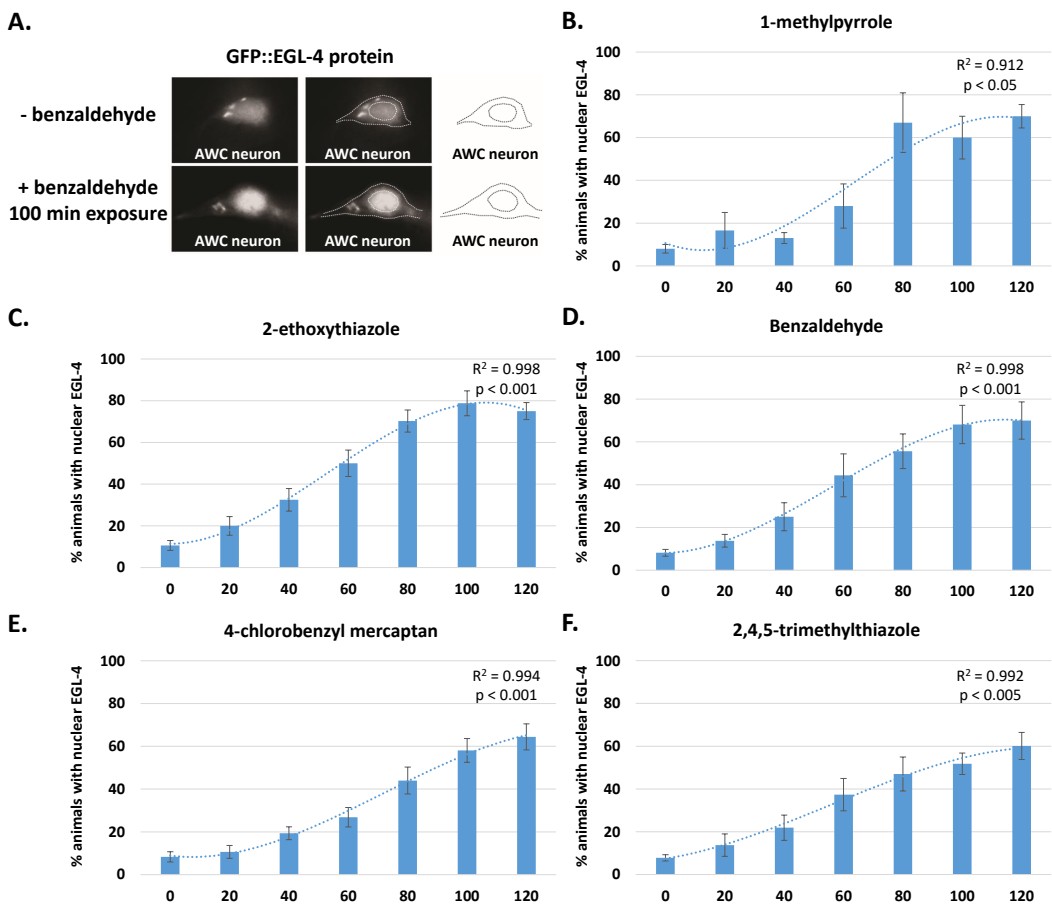

**Figure 4  GFP::EGL-4 nuclear accumulation in early and late trending odors.** Animals were exposed to odor for indicated time and cytoplasmic/nuclear GFP::EGL-4 localization was observed. (A) GFP::EGL-4 expression in the AWC neuron of a worm exposed to buffer only (A, B) and a worm exposed to the odor benzaldehyde for 100 min (E, F). Outlines indicate the boundaries of the AWC neuron cell body and nucleus. (B–F) Graphs indicate percent animals with nuclear EGL-4 over time for early-trending odors (B, C, D) and late-trending odors (E, F). A polynomial regression was calculated and regression curves are indicated on each graph. Regression correlation coefficient and *p*-values calculated by ANOVA analysis are indicated. See Methods for individual regression equations. Error bars indicate standard error.

them into the early-trending odors benzaldehyde, 2-ethoxythiazole, 1-methylpyrrole and the late-trending odors 2,4,5-TMT and 4-CB. We find that these three early-trending odors induce a 50% change in behavior after an average of 83.3 min whereas the two late-trending odors induced a 50% change at 101.8 min, significantly later than early-trending odors (Fig. 5). We then exposed worms to early and late-trending odors and observed EGL-4 localization in the AWC neuron every 20 min for 2 h (Figs. 4B–4F). Whereas the odor butanone is sensed by one of the AWC neuron pair and EGL-4 nuclear translocation is induced in only one of the AWC neurons, all the odors tested here could induce EGL-4 translocation in both AWC neurons indicating that they are likely sensed by both AWC neurons. The maximal percent of animals with nuclear EGL-4 ranged from 60% to almost 80% for each odor (Figs. 4B–4F). However, neither longer odor incubation nor increasing

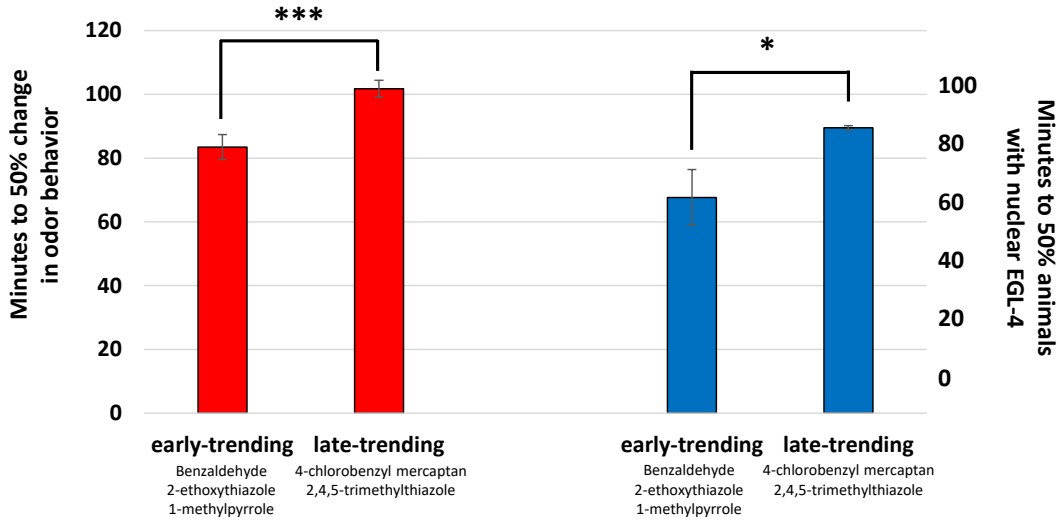

**Figure 5** **Timing of behavior change and EGL-4 nuclear accumulation in response to early and late-trending odors.** The time to 50% behavior change to early-trending and late-trending odors is indicated by the red bars, and the time to 50% of animals with nuclear EGL-4 after exposure to early-trending and late-trending odors is indicated by the blue bars. Error bars indicate standard deviation. Significance was determined by student's $T$-test analysis. * indicates $p < 0.05$, *** indicates $p < 0.001$.

odor concentration could substantially increase maximal nuclear EGL-4 for 4-chlorobenzyl mercaptan or 2,4,5-trimethylthiazole (Figs. S7 and S8).

The temporal dynamic of EGL-4 nuclear localization in response to the early- and late-trending odors seems to vary depending on the odor. To analyze this in detail, we calculated a regression for the time curves for each odor (Figs. 4B–4F), and then estimated the time at which 50% EGL-4 nuclear localization was reached for each odor (Fig. 5). We found that EGL-4 nuclear localization occurred much faster in the early trending odors (58 min, 70 min, 75 min for 2-ethoxythiazole, benzaldhehyde and 2-methylpyrrole, respectively), but significantly later in the late-trending odors (90 min and 89 min for 4-CB and 2,4,5-TMT, respectively). Thus, we saw a correlation between the timing of behavior change and EGL-4 nuclear translocation (Fig. 5). Taken together, the timing of odor behavior changes for AWC-sensed odors may be regulated by EGL-4 sub-cellular localization and aversive learning mechanisms.

## DISCUSSION

Establishing the real-time behavior assay was the key to comprehensively testing temporal dynamics of adaptation and aversive learning to multiple odors. This assay has several advantages to the standard assay that has been valuable in understanding odor behavior in *C. elegans*. Firstly, tracking a single population over time rather than independent populations at each time point has benefits. For instance, we can observe actual changes in the behavior of animals over time and save time and resources tracking one population rather than 13 populations over the 120 min assay. In addition, the real-time assay is a better simulation of odor behaviors in natural habitats than the standard assay. Our group

previously showed that production of the odor diacetyl in rotting fruit attracts *C. elegans* (*Choi et al., 2016*). The new assay allows worms to freely move on a large field and alter their behavior towards the odor over time more similar to natural habitats.

Although tracking a single population over time has advantages, this results in each data point being dependent on the previous data point. This could lead to a decrease in resolution in data over time that is not a problem in the standard assay. Due to this concern we tested whether different behavioral states in *C. elegans* mutants can be resolved in the real-time assay (Fig. 2). We found that short-term odor adaptation defects were observed in *cng-3* mutants in the real-time assay as has been observed in the standard assay (*O'Halloran et al., 2017*). Still, the standard assay has stronger behavior resolving power than our new assay particularly in identifying behavior mutants. The real-time assay cannot replace the standard behavior assay but can supplement it with an ability to observe temporal dynamics on a large scale and also simulate natural odor behaviors.

Although *C. elegans* odor behaviors have been well characterized for decades, some confusion remains about the terminology of these behaviors. Recently, it was shown that the AWC neuron decreases its responsiveness to the odor butanone after a 1-hour persistent exposure (*Cho et al., 2016*). Thus, *C. elegans* displays both behavioral and neuronal adaptation to butanone. Worms also associate butanone with food deprivation after a 1-hour persistent exposure in the absence of food, and show a learned aversion to the odor independent of adaptation (*Cho et al., 2016*). Therefore, *C. elegans* can adapt to and also form an aversive memory towards AWC-sensed odors after 60 min that is long lasting (*Lee et al., 2010*) and is lost in *adp-1* mutants. Worms also display a temporary weakened attraction to AWC-sensed odors after a 30 min exposure that is dependent on components of the cyclic nucleotide gated channel TAX-2 and CNG-3, and has been termed "short-term adaptation" (*L'Etoile et al., 2002*; *Lee et al., 2010*; *O'Halloran et al., 2017*). It is unknown, however, if this short-term adaptation is related to the neuronal adaptation displayed in the AWC neuron after a 1 h butanone exposure. In any case, we have used the terms "adaptation" to refer to *cng-3*-dependent changes in odor behavior, and "aversive learning" to refer to nuclear EGL-4 and *adp-1*-dependent changes in odor behavior.

To facilitate this study, we used a subset of 12 odors to test worm behavior in the rea-time odor behavior assay. Using the real-time assay we were able to identify odors in which adaptation or aversive learning occurs early in *C. elegans* such as 1-methylpyrrole, and odors in which learning occurs late such as 4-chlorobenzyl mercaptan. Since aversive learning to AWC-sensed odors requires EGL-4 nuclear translocation, we exposed worms to early-trending and late-trending odors and observed EGL-4 localization over time. We found a correlation between the timing of behavior change and nuclear accumulation, suggesting that the changes in odor behavior over time are mediated by aversive learning mechanisms. Although our experiments confirm that aversive learning is occurring in both early and late-trending odors, we cannot rule out the role that odor adaptation may be playing in the temporal aspects of odor behavior for the odors tested. Likely, the behavior patterns we observe in the real-time behavior assay are a combination of adaptation and aversive learning mechanisms.

How then do specific odors regulate the timing of EGL-4 nuclear translocation? Aversive learning in the AWC neuron is dependent on the coincident detection of food deprivation that is mediated by insulin signaling from the AIA neuron (*Cho et al., 2016*). Insulin signals from the AIA activate the PI3 kinase AGE-1 in the AWC neuron to promote EGL-4 nuclear accumulation. However, food deprivation is stable from odor to odor in our experiments, thus AIA neuron-dependent insulin signals likely cannot account for the differences observed in odor-dependent EGL-4 nuclear translocation. Within the AWC neuron, the G-protein ODR-3 and cGMP levels have both been shown to regulate nuclear EGL-4 localization (*O'Halloran et al., 2009*; *O'Halloran et al., 2012*). Specific odors, and possibly specific G-protein coupled olfactory receptors, may target ODR-3 and/or cGMP machinery which includes the guanylyl cyclases ODR-1 and DAF-11 and AWC-specific phosphodiesterases to regulate the timing of EGL-4 nuclear localization (*Birnby et al., 2000*; *L'Etoile & Bargmann, 2000*; *Roayaie et al., 1998*). Further experiments that identify new olfactory receptors in the AWC neuron will elucidate the specific mechanisms.

One possible explanation of the differences we observe in time-dependent behavior change is that odor attraction is concentration-dependent, and adaptation and learning here may also depend on concentration. For this reason, we tested several dilutions for most of the odors analyzed in the real-time behavior assay (Figs. S7 and S8), and finally chose the optimal dilution for each odor in which behavior change was greatest over the entire experiment. For odors that have been previously tested for attraction such as benzaldehyde and butanone, the concentrations used in our study were slightly lower than those used in standard chemotaxis assays (*Bargmann, Hartwieg & Horvitz, 1993*). In addition to odor concentration, worms inherently display varying levels of initial attractiveness to the odor (Table 1, Fig. 3), and this difference could affect temporal-dependent changes in behavior. However, we showed that behavior differences between early and late-trending odors did not necessarily correlate with initial AI to those odors (Fig. S5), and differences in initial AI to the same odors did not significantly impact temporal changes in behavior (Fig. S6). A previous study has shown that even undiluted benzaldehyde and isoamyl alcohol induced normal aversive learning (*Colbert & Bargmann, 1995*). In this way, we observed that the collective change in attraction over the whole assay was similar for 12 of 13 odors tested. Despite overall similarities at the optimal concentrations, temporal patterns of behavior change were still varied among the odors. However, a more thorough analysis will need to be conducted in order to understand the full role of odor concentration on the temporal aspects of odor behavior.

We also assessed behavior change to the AWA-sensed odor diacetyl and found that change occurs faster and more consistently than AWC-sensed odors. In addition, the odor 2,4,5-trimethylthiazole is sensed by both AWA and AWC neurons. Interestingly, we found that 2,4,5-TMT induced EGL-4 nuclear accumulation in the AWC neuron, and late behavior changes was correlated to late EGL-4 translocation. Further testing with AWA-sensed odors such as pyrazine (*Bargmann, Hartwieg & Horvitz, 1993*) and butyric acid and isobutyric acid (Table 1) may reveal whether fundamentally different mechanisms of odor memory formation exist between the AWA and AWC neurons.

In this study, we investigated *C. elegans* response to many odors, including odors that have not been tested on *C. elegans* before, and identified the sensory neuron responsible for each odor attraction. In addition, we tested the memory formation towards 13 of the odors over time. Such inherent behaviors towards the odors indicates that these odors may be ecologically relevant cues for *C. elegans* in nature. For instance, 2-isobutylthiazole is a major component of antelope pheromone (*Burger et al., 1988*), 2-methylpyrazine is the main volatile component of grape vinegar (*Pinu et al., 2016*), and butyric acid is produced by microbial fermentation (*Pasteur, 1861*). Indeed, studies have shown ecologically relevant relationships between *C. elegans* and diacetyl (*Choi et al., 2016*), methyl 3-methyl-2-butenoate (*Hsueh et al., 2017*), and 2-heptanone (*Zhang et al., 2016*), and mammalian predator–prey relationships between 2,4,5-TMT, a component of fox feces, and rodents (*Vernet-Maury, 1980*). Finally, we found that memory does not form towards cyclohexylethanol. The ecological implications of a persistent attraction to this odor is unknown. Cyclohexylethanol derivatives were found in a plant root extract widely used in Chinese folk medicine (*Huang et al., 2009*), but relationships with nematodes need to be further investigated. We hope that this study can be a platform for more studies into natural odor behaviors.

## CONCLUSION

Dynamics of learning may vary depending on the cue that is learned. Since *C. elegans* can learn to associate odor deprivation to a multitude of odor cues, we devised a new behavioral assay to efficiently examine the temporal dynamics of adaptation and learning to over a dozen odors. Using the real-time behavior assay, we found behavior changes to certain odors occur faster than others. Finally, we investigated the cellular basis for this difference and found odors that are learned early also cause EGL-4 to translocate into the nucleus earlier than odors that are learned late.

## ACKNOWLEDGEMENTS

Strains were provided by the Caenorhabditis Genetic Center. The project was carried out as part of the 2017 Undergraduate Research Program from the Korea Foundation for the Advancement of Science & Creativity (KOFAC).

### Funding

This work was supported by the National Research Foundation of Korea Grant 2016R1D1A1B03932745 to Jin I. Lee and Grant 2016R1C1B1011269 and 2017R1A5A2015369 to Kyoung-hye Yoon. The funders had no role in study design, data collection and analysis, decision to publish, or preparation of the manuscript.

### Grant Disclosures

The following grant information was disclosed by the authors:

National Research Foundation of Korea Grant: 2016R1D1A1B03932745, 2016R1C1B1011269, 2017R1A5A2015369.

## Competing Interests

The authors declare there are no competing interests.

## Author Contributions

- Jae Im Choi, Hee Kyung Lee, Hae Su Kim and So Young Park performed the experiments, authored or reviewed drafts of the paper, approved the final draft.
- Tong Young Lee analyzed the data, prepared figures and/or tables, authored or reviewed drafts of the paper, approved the final draft.
- Kyoung-hye Yoon conceived and designed the experiments, analyzed the data, contributed reagents/materials/analysis tools, prepared figures and/or tables, authored or reviewed drafts of the paper, approved the final draft.
- Jin I. Lee conceived and designed the experiments, performed the experiments, analyzed the data, contributed reagents/materials/analysis tools, prepared figures and/or tables, authored or reviewed drafts of the paper, approved the final draft.

## Data Availability

The raw data are provided as Supplemental Files.

## Supplemental Information

Supplemental information for this article can be found online at http://dx.doi.org/10.7717/peerj.4956#supplemental-information.

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
