# Peer review of "Odor-dependent temporal dynamics in Caenorhabitis elegans adaptation and aversive learning behavior"

_PeerJ, doi:10.7717/peerj.4956_

## Round 0.1 · original submission · Major Revisions

Overall, the piece represents interesting primary research. The reviewers and I agree that it was well-prepared and contributes to the primary literature while building on previous research. I am particularly taken with the idea of trying to create a natural habitat to understand normal behavior. The literature review cites appropriate literature to frame the problem and discussion. Ultimately, the overall tone of the reviews and my reaction are positive. The reason we are asking for major revisions is the framing of the piece and the analytical component but we do not feel these are fundamental flaws—simply a matter of clearer communication and consistency in reporting.

The first issue is that we note the paper to be about odor adaptation, not learning. There is some confusion over what is meant by learning as the outcomes here are not the classical understanding of the short-term and long-term memory but declines in attraction and repulsion. The changes taking place cannot be attributed strictly to receptor functioning; repulsion is a neural mechanism (or, perhaps in worms, as one reviewer notes—whole system output) that may occur even if the receptor is active. I wonder if the simulation of a more natural habitat where the worms are allowed to follow their noses so to speak, is how the notion of learning rather than adaptation is introduced…? A revision should be very careful with terms and either introduce what is meant by learning (if adaptation, maybe simpler to use the term, given the literature cited) or to walk us through how this is actually about learning. The second issue is the statistical reporting and analysis. There is no analytical evidence for effects of odors, which is needed given the variation in parameters. In particular, note comments regarding the missing information needed for basic statistical reporting, labeling of raw data, consistency in figure labeling and formatting, inclusion of raw data (including the larger 150 odor dataset).

If you feel that these changes are acceptable, I strongly encourage a resubmission.

·

Basic reporting

This article is nicely reported. Language throughout is clear and professional. The introduction and background sections reference relevant literature and give appropriate context to the work done. The structure conforms to PeerJ standards. Appropriate raw data is supplied, although additional labeling would be helpful for clarity. Figures are relevant and well labeled, with a few improvements suggested below.

1. Odor memory in this paper is odor adaptation, which is best given clarification (e.g., as in Cho et. al 2016) to distinguish from other forms
2. To improve consistency and facilitate comparisons, all figure data should be presented as uniformly as possible, including axis scales, axis label positions, start times on x-axis, etc. For example, in Figure 4, methylpyrrole graphs begin at 0 minutes whereas others begin at 20 min.
3. Similarly, organizing graphs in Figs. 3 and 4 in the same positions (for those odors present in both) would help comparison.
4. In Table 1, benzyl proprionate and butyric acid have equal responses in AWA- and AWC- strains, yet the conclusions of which neurons are required to detect each differ. Please clarify.

Experimental design

This work represents original primary research within the scope of PeerJ. The research question answered was well-defined and meaningful, potentially providing useful insight into the timing of odor memory in a set of attractive odors sensed by AWC by timing of EGL-4 nuclear accumulation. The experimental work duplicated or modified existing techniques and were performed to accepted ethical standards. Methods were mostly described in sufficient detail to replicate results.

1. For quantification of EGL-4 translocation, additional method detail would be helpful to understand data variability and confidence. The metric of % animals with nuclear EGL-4 would seem to require a binary assessment of each individual (i.e. yes/no for nuclear expression). But is translocation sudden or gradual? How ambiguous or clearly distinguished were animals with accumulation of EGL-4? Example images demonstrating EGL-4 accumulation and the counting method for accumulation assessment will improve clarity for Figure 4.

Validity of the findings

Data appears well-gathered, with appropriate controls taken; however, substantial statistical concerns are present as described below. As a result, there are some unclear links to the conclusions stated in the article. Further, both late-trending odors appeared saturated in early behavior tests, possible affecting conclusions.

1. Statistical analysis is missing from the primary conclusion of this paper: the correlation of temporal dynamics of odor memory with dynamics of nuclear accumulation of EGL-4. The speed of memory formation is quantified as the time to reach 50% of the total change in attraction index (AI) over 120 minutes, but since there is variability in AI at each time point, there is also uncertainty in the 50% change time. This uncertainty should be reported, such as by standard deviation (SD) or 95% confidence interval. Similarly, timing of nuclear accumulation is quantified as the time to reach translocation in 50% of animals from a curve fit to discrete time points, each with measurement variability. Therefore, SD or 95% confidence intervals for these values should be reported. These data are summarized in Table 2, which should also list the SD’s or 95% confidence intervals, with appropriate significant figures. Finally, the correlation of behavior change to EGL-4 localization should be made with statistical comparisons taking into account these uncertainties.
2. Regression data for Figure 4 is overfit to available data (a 4th-order polynomial to 6-7 data points). The data appears appropriate for a sigmoid function fit, which would contain a regression parameter directly related to the inflection point, a good metric of timing.
3. The authors rightfully concern themselves with odor concentration, which can affect odor dynamics, adaptation rates, which neurons respond, etc. Here, test concentrations were chosen by optimizing for greatest behavior change over 2 hours. However, it appeared that some odors (e.g. 4-chlorobenzyl mercaptan and 2,4,5-trimethylthiazole) had initially high and stable AI (with low variability and small error bars compared to other data points), indicating possible saturated attraction. Could this effect alone delay the behavior change and make these odors appear late-trending? Instead, normalizing concentrations for the same baseline attraction, such as concentrations resulting in a non-saturated value of AI around 0.75, rather than maximal behavioral change, may better elucidate differences in odor memory and attraction. This would also allow for comparison figures overlaying odors to demonstrate behavioral shifts that may correlate to EGL-4 accumulation. Alternatively, observing similar late-trending changes at lower concentrations of these odors would lend confidence to the conclusions that EGL-4 translocation timing determines odor memory timing.
4. The authors mention initial testing of 150 odors, from which they identified dozens of attractive odors (it would be helpful to include all these initial data). These were further narrowed down to a subset of AWC-detected odors, and one AWA-detected odor for testing with behavior, from 5 odors were chosen for studying EGL-4 accumulation. What was the rationale for including or excluding data? For example, behavior change to butanone seems one of the most early-trending (Fig. 3), compared with 4-chlorobenzyl mercaptan (late-trending), but data for EGL-4 localization isn’t included here.

Additional comments

This article investigates numerous attractive odors to assess odor memory and timing in the AWC neuron, testing an interesting hypothesis that timing of memory formation in AWC follows timing of EGL-4 accumulation in the nucleus. Through their screen, the authors identify several odors showing high attraction sensed by AWC, including many previously untested, and demonstrate through a dynamic chemotaxis assay format that C. elegans form an odor memory through adaptation, resulting in decreased attraction over time. While the assays are well-stated and performed, variability in raw data and in timing metrics call for statistical analyses to determine whether the odor memory timing correlates properly to the EGL-4 accumulation timing. If after further statistical analysis this correlation holds, the resulting story yields useful insight into the mechanisms of odor memory regulation and timing.

Reviewer 2 ·

Basic reporting

The article is well-written and generally clear, although I have a major problem with how the experiments are framed in terms of “memory” (see 3 and General comments). The main difficulty I found with the basic reporting is how the real-time odour behaviour assay worked – I do not see how it tests different “memory” formation time-courses. From what I can understand, you allow a population of worms to migrate and note their position every 10 minutes over a 120 minute time period (figs 2A and 2B). You then determine whether different odours have different time courses in their responses (using the 50% change as a key criterion), which you nicely correlate with EGL-4 nuclear localisation. You argue that this is better than looking at independent groups, and is certainly easier to study. My main concern here (apart from the fact that this is poorly explained) is that this has nothing to do with ‘memory’. If you described this simply as changes in odour response over time (or adaptation), it would be so much clearer.

Experimental design

I cannot find how many replicates you carried out for each R-TOB assay. This should be given in both the text and the figure legend. This also affects your statistical tests (such as they are) and your use of standard errors to demonstrate variability. You claim that your R-TOB is better than the usual assay it reduces experimental variability between populations – you could show this by using a box and whisker plot. It is not clear to me how you compared the effects of different odours on your 50% change criterion, or how you compared slopes produced by different odours. Some kind of statistical analysis needs to be carried out here.

Validity of the findings

I find it hard to understand why the authors have chosen to frame their work in terms of learning. All of the previous articles they cite refer, correctly in my view, to adaptation to describe the phenomenon studied here. There is no operant or classical conditioning taking place in these experiments (these are generally taken to be ‘learning’), for there is no reinforcer. The same applies to ‘short term’ and ‘long term memory, which are not, as far as I can see, appropriate terms, for they are not linked to the normal use of these terms – here they refer instead to a decline in attraction and then to repulsion (if I have understood things correctly).

As far as I can understand it, this study has explored changes in olfactory response over time, during a continuous stimulation. There are two prime ways that this could take place – either on the membrane, in terms of changes to receptor/second messenger activity, or at the synaptic level, whereby the cell may no longer be transmitting a response. The transformation of attraction to repulsion may not be a direct consequence of the receptor cells that you are observing and manipulating, but instead to the output of the whole system following continuous stimulation – they are certainly not the equivalent of LTM and STM.

This phenomenon more appropriately termed adaptation and I feel quite strongly that the whole article needs rewriting/reframing along these lines. If the authors have a good case to maintain their use of ‘memory’, they must make it explicitly, explaining why ‘adaptation’ is no longer the appropriate framework for this kind of study. It may be that I have missed some recent developments in the worm world, but it seems to me that the authors have mistakenly over-interpreted here. Adaptation as a basic sensory process is just as interesting as memory formation! The explanations of what has taken place here, in terms of intracellular activity are potentially important, at a far more fundamental level in terms of a) sensory neuron function and b) the activity of a neuronal network governing behaviour in a model organism. This is the way that the article should be framed, I think.

Additional comments

I found this article much easier to understand once I realised it wasn't about memory as I understand it, but instead about adaptation. I think the correlation of behavioural changes and EGL-4 nuclear localisation is really nice, and if you carry out the kind of mainly literary changes I am suggesting (assuming the Editor agrees), I think this will be a really useful contribution.

---

## Round 0.2 · Minor Revisions

Thank you very much for your resubmission of this interesting paper. I am particularly taken with experiments that attempt to create a more natural setting and I like your careful discussion of the advantages of your assay and the advantages of the standard assay as having combined power to test behavior. Ultimately, I think the protocols, the care with which they were carried out, and the findings constitute a very important contribution. I would say there are two main areas which require just a small amount of editorial work before the manuscript can be accepted for publication.

First: The reviewers and I agree that there is continued confusion over what behavior is adaptive and what behavior is aversive learning. On Line 51 you clearly state that the observed behavior is aversive learning but continue to include adaptation throughout the paper as a auxiliary term. As one reviewer notes, the literature from which this research sprang is confusing. This is an excellent opportunity for your lab to clarify what specifically is observed in these assays. My view, from an evolutionary background, is that adaptation cannot be claimed unless you conduct additional experiments that show a consistent change in behavior over time in these animals. Aversive learning, as described in the paper, appears to be regulated by neuronal processes and could be reversed, I imagine, in an assay that provides the previously expected reward (food) in association with the odor. If experiments cannot identify one of two interpretations for the same behavior then we cannot say whether the behavior is learned or adaptive. Thus, the discussion of what you have found must be revised to consider the limits of what we can infer from these results. Please refer to the specific reviewer comments in this regard.

Second: The paper needs reorganization. Much of the material presented in the Results section is part of the introduction and methods that are needed for the reader to understand how the experiment is set up. Only the results of the experiments belong in the results section. I note the following incorrect placements using the line numbers from the PDF document:
130-160: This section justifies the use of the odors and should be moved to the methods section for Behavior Assays inserted at Line 73 before the sentence “All odors…”. Then form a new paragraph with after the dilution sentence.
157-182: This section is a better version of what is currently at Lines 53-59 so work these two sections together into one and keep them in the introduction.
190-213: This section appears to be justification and background information regarding the assay proposed and should be moved earlier into the introduction with lines 157-182
A few other minor points to clarify:
Line 80: You refer to AI without explaining what is. Replace AI with Attraction Index (AI). For all other instance of attraction index, use only AI (see Line 104 where you spell it out and line 445 where it is actually first defined).
266-68: Here you seem to define adaptation as not involved nuclear localization and aversive learning as involving nuclear localization (of EGL-4). Is this correct? If so, this definition and a justification for it should be presented very early in the introduction.
271-2: The use of early-trending and late-trending describes how early a behavior occurs in the assay, I think. Are these standard terms. If not, a little clarification of your short-hand would be helpful.
308 and 353: You seem to be presenting new data or discussing data not previously disclosed in results starting with these two lines. Be sure to carefully review the discussion and place any primary results (even if not reported with data) in the results section.

·

Basic reporting

Revision addresses adaptation terminology and improves consistency of data figures.

Experimental design

The revised figure panel clearly describes the EGL-4 translocation assay.

Validity of the findings

The revision addresses some of the prior review comments. However, it doesn’t address the main concern regarding the authors’ primary conclusion that differences in adaptation and EGL-4 translocation timing are odor-specific. Both “late-trending” odors (4-CM and TMT) had very high initial AI’s which did not change much over the first 70-80 min (i.e., the measurement appeared saturated). Therefore, the time to reach 50% of the change between 30 and 120 minutes is necessarily larger than for an odor that shows an early AI decline (assuming similar rates of AI decline across odors, which appears to be the case). It is possible (or likely) that the same odors would have lower initial AI's at lower concentration. If so, would AI's be stable over the first 70-80min, like at the higher concentrations tested, or decline sooner, like the "early trending" odors?

In short, since only one concentration was chosen for each odor, it is unclear that differences in adaptation timing are inherent to specific odors instead of concentration-dependent. (For example, is there a significant correlation between initial AI and adaptation timing across all odors at the concentrations tested?)

Regardless of the cause, the authors show nicely the correlation between timing of behavior adaptation and EGL-4 translocation. The specific claim that odors are inherently “early-trending” or “late-trending” should only be made if the alternative that odor concentration determines early- or late-timing is ruled out, such as by testing 4-CM and TMT at lower concentrations. Alternatively, the discussion should include both possible interpretations of the data.

Also, as indicated previously, a monotonic function such as a sigmoid fit, would be better than a polynomial fit for translocation data.

Additional comments

The revised article is an improvement and clearly demonstrates a correlation between delayed localization of EGL-4 and delayed adaptation in chemotaxis behavior. However, the conclusion that these timing differences are inherent to specific odors isn’t fully supported, as they could also depend on concentration (e.g. one odor could be early-trending at one concentration and late-trending at another). Evidence should be presented toward one interpretation, or both should be included equally.

Reviewer 2 ·

Basic reporting

.

Experimental design

.

Validity of the findings

.

Additional comments

This is much better, and I am beginning to understand what the authors intended. The absence of food is clearly a negative reinforcer for these very hungry animals.

However, despite the role of insulin signalling in the decline in attraction (p3/lines 11-13) you cannot simply conclude that this is learning. As you now state, adaptation is also taking place – teasing apart the two effects is difficult. Although I accept that a) confusing terms have been used in the literature in the past and b) the adp mutant is apparently specific to learning, I do not understand how your real time assay only measures learning, and not adaptation. You have a paragraph on p11 lines 4-19, which I have now read three times and do not understand how “short-term adaptation and aversive learning can be resolved in the real-time behavior assay” in non-mutant worms, which are mainly what you study in the next section.

If I am correct, and you cannot in fact separate them, then you need to do some further editorial work, including in the title and the abstract, clarifiying this wherever you make claims about memory/learning. The fact that the literature is confused is not a justification for perpetuating that confusion! You really need to explain this very clearly, because at the moment, although you have nuanced the rest of the article, this section, and indeed the title, are still focusing on learning when it seems to me that you have two entwined phenotypes – learning and adaptation.

If I am mistaken, I am happy to be corrected, but you should consider that other readers may be similarly confused, and some editorial clarification might be advised.

You therefore need to make sure that you don’t simply write about aversive learning, so please add ‘adaptation and’ before ‘aversive learning’ at 7/13, 8/17, 9/3

Also:
8/22 replace ‘molecular pathways involved in associative learning’ by ‘the molecular pathways involved’
9/6 delete ‘odor memory behaviour
10/14 replace ‘aversive learning’ by ‘change in attraction’

---

## Round 0.3 · accepted · Accept

Thank you for your careful attention to the peer review process and producing a strong manuscript with interesting findings.

#